# Review on the Use of Superconducting Bulks for Magnetic Screening in Electrical Machines for Aircraft Applications

**DOI:** 10.3390/ma14112847

**Published:** 2021-05-26

**Authors:** Rémi Dorget, Quentin Nouailhetas, Alexandre Colle, Kévin Berger, Kimiaki Sudo, Sabrina Ayat, Jean Lévêque, Michael Rudolf Koblischka, Naomichi Sakai, Tetsuo Oka, Bruno Douine

**Affiliations:** 1Groupe de Recherche en Energie Electrique de Nancy, Université de Lorraine, GREEN, 54000 Nancy, France; remi.dorget@univ-lorraine.fr (R.D.); kevin.berger@univ-lorraine.fr (K.B.); jean.leveque@univ-lorraine.fr (J.L.); bruno.douine@univ-lorraine.fr (B.D.); 2Safran Tech, Electrical & Electronic Systems Research Group, Rue des Jeunes Bois, Châteaufort, 78114 Magny-Les-Hameaux, France; sabrina.ayat@safrangroup.com; 3Experimentalphysik, Saarland University, P.O. Box 151150, 66041 Saarbrücken, Germany; m.koblischka@gmail.com; 4Airbus UpNext, 31300 Toulouse, France; alexandrecolle57@gmail.com; 5Shibaura Institute of Technology, Tokyo 135-8548, Japan; ac16052@shibaura-it.ac.jp (K.S.); nsakai@shibaura-it.ac.jp (N.S.); superoka49@gmail.com (T.O.)

**Keywords:** superconducting bulk, superconducting electrical machine, flux modulation machine

## Abstract

High-Temperature Superconductors (HTS) considerably accelerate the development of superconducting machines for electrical engineering applications such as fully electrical aircraft. This present contribution is an overview of different superconducting materials that can be used as magnetic screens for the inductor of high specific power electrical machines. The impact of the material properties, such as the critical temperature ($T_{c}$) and the critical current density ($J_{c}$), on the machine performances is evaluated. In addition, the relevance to flux modulation machines of different HTS bulk synthesis methods are addressed.

## 1. Introduction

The development of High-Temperature Superconductors (HTSs) such as the cuprate family with the REBaCuO superconductors (Rare Earth elements (RE)), the Iron-Based Superconductors (IBS) with the Pnitide and Chalcogen families and the MgB${}_{2}$ compound allow a quick evolution of superconducting devices for many applications. Among them, lead by ecological, economical, and practical needs, the electrical machines associated with superconducting materials such as electromagnet [1], permanent magnets [2,3,4,5], or windings [6,7] look increasingly more attractive for future transports such as electrical ships [8], aircraft [9,10,11], and levitation trains [12] for the high specific power potential of HTS machines. Indeed, for the aviation sector in particular, the growing need for electrical power on board requires the development of electrical machines with a target power-to-weight ratio of 20kW/kg for MW class machines [4,13,14].

Several topologies of HTS synchronous machines are considered promising such as rotor-wound machines [15,16]; superconducting “permanent magnet” machines, made from HTS bulks [17] or HTS stacks of tapes [18]; and flux modulation machines [19,20,21,22,23]. The last topology, as illustrated in Figure 1 in its axial flux version, is composed by a static solenoid coil made of superconducting tapes which produce a DC magnetic field. The second element of the inductor consists of superconducting bulks located on the rotor; these pellets are used in the mixed state, thus the application of the coil magnetic field will induce currents in the material, which will screen the coil field. The flux density behind the bulks is therefore reduced, and there is a magnetic flux modulation in the air gap. Eventually, this variations will induce an electromotive force in the three phase windings located on both sides of the rotor. This topology does not requires any slip rings contrary to rotor-wound machines and the magnetization can be controlled, unlike superconducting “permanent magnet” machines while showing promising performances. Indeed, flux modulations machines reach high specific powers by having a large magnetic field in the air gap (>2 T), whereas conventional machine are limited by the iron magnetic saturation. Thus, it is possible to reach high power to mass ratio without increasing the rotational speed. In contrast to aircraft conventional machines, which tend to increase in speed and require the use of a mechanical gearbox [24], the expected operating speed for flux modulation motors is only a few thousand rpm.

In order to properly screen a large magnetic field, the material used for the rotor must be a efficient superconductor. Furthermore, as liquid hydrogen is likely to become the main coolant for aircraft applications [11], the working temperature of the machine must be approximately 20 K. Therefore, to ensure a comfortable margin, a critical temperature $T_{c}$ above 30K is needed for the potential candidates.

Along this paper, we will analyze and evaluate the performance as well as the relevancy of the following bulk superconducting materials for use in a reliable industrial flux modulation machine for aircraft applications:The Pnictide and Chalcogen families of Iron-based superconductors (IBS) [25,26,27,28] with $T_{c}$ as high as 46K for the NaFeSe [29,30] and 56K for the Sr ${}_{0.5}$ Sm ${}_{0.5}$ FeAsF [31] compound. The IBS also show very high critical magnetic field ($H_{c2}$) [32] and promising critical current density ($J_{c}$) at liquid hydrogen temperature [33].The REBaCuO materials [34] with the so-called YBaCuO [35] and GdBaCuO [36], which are the most widely used for electrical engineering applications because of their really high superconducting properties [37,38].The MgB ${}_{2}$ compound [39] with a $T_{c}$ of 39K, an acceptable $H_{c2}$ and a very high $J_{c}$ [40,41] has a good potential as trapped field permanent magnet [42,43].

The required superconducting properties, i.e., the critical current density $J_{c}$, for that type of machine will be determined using simulation tools in a first section. In a second section, the existing synthesis and preparation techniques for each materials and their relevancy for custom and mass production will be highlighted.

## 2. Critical Current Density and Magnetic Shielding Properties

Naturally, the critical current density $J_{c}$ of HTS materials has an impact on a flux modulation machine performances. To study the $J_{c}$ influence on the flux modulation quality, we made a numerical 2D axisymetric model of the bulk using the commercial software COMSOL Multiphysics. The parameter values of this model are listed in the Table 1. The Maxwell equations are expressed by the H-formulation linking the single component of the electric field along the θ-direction $E_{\theta }\left(r,z\right)$ and the two components of the magnetic field along the *r* and *z* directions $H_{r}\left(r,z\right)$ and $H_{z}\left(r,z\right)$, respectively:(1)$$\mu_{0}\frac{\partial H_{r}\left(r,z\right)}{\partial t}-\frac{\partial E_{\theta }\left(r,z\right)}{\partial z}=0$$
(2)$$\mu_{0}\frac{\partial H_{z}\left(r,z\right)}{\partial t}+\frac{\partial E_{\theta }\left(r,z\right)}{\partial r}+\frac{E_{\theta }\left(r,z\right)}{r}=0$$

The magnetic field is linked to the current density single component $J_{\theta }\left(r,z\right)$ through the Maxwell–Ampere equation:(3)$$J_{\theta }\left(r,z\right)=\frac{\partial H_{r}\left(r,z\right)}{\partial z}-\frac{\partial H_{z}\left(r,z\right)}{\partial r}$$

The electromagnetic behavior of the bulk is modeled through an E−J law with a constant $J_{c}$ [44]:(4)$$E_{\theta }\left(r,z\right)=\frac{E_{c}}{J_{c}}{\left(\frac{||J_{\theta }\left(r,z\right)||}{J_{c}}\right)}^{n-1}J_{\theta }\left(r,z\right)$$

The magnetic field applied by the coil is modeled as an homogeneous magnetic field applied in the *z*-direction. The simulation is time-dependent and consists of two steps: First, the applied field ramp rise from 0T to the desired value $\mu_{0}H_{a}$ in 100s. The applied field is then kept constant for 100s to reach the steady state at which the the flux modulation is observed 3mm behind the bulk. Eventually, we evaluate the performances by calculating the amplitude of the first space harmonic of the field variation $B_{1}$, which is proportional to the torque produced in an electrical machine and is calculated from
(5)$$B_{1}=\frac{1}{R}\int_{0}^{2R}\mu_{0}H_{z}\left(r,\frac{h}{2}+3\;\mathrm{mm}\right)\cos\left(\frac{2\pi r}{4R}\right)dr$$

Figure 2 shows the variation of $B_{1}$ versus the $J_{c}$ for different values of applied magnetic field. Although the critical current density has a clear influence on the machine power, one can observe that above a certain threshold, the flux modulation amplitude does not increase anymore. Thus, a bulk with a critical current equal to $2000\;A/{\mathrm{mm}}^{2}$ will provide performances close to a perfectly diamagnetic bulk which corresponds to an infinite critical current.

To properly understand this last result, one must analyze the behavior of the magnetic field and current in the bulk while subjected to the coil field. Figure 3 shows the 3-D current distribution in the bulk while subjected to 3 T for $J_{c}$ equal to $500\;A/{\mathrm{mm}}^{2}$ and $2000\;A/{\mathrm{mm}}^{2}$. It can be seen that the current is penetrating the bulk deeply at $500\;A/{\mathrm{mm}}^{2}$ while remaining close to the surface above $2000\;A/{\mathrm{mm}}^{2}$. As shown on Figure 4, the parts of the bulks penetrated by the current are those where the field is not properly screened. Therefore, increasing $J_{c}$ will decrease the penetration depth of the bulk, which explains the strong gains in performance when going from complete penetration to a case where the current only flows at the extremities. However, above $2000\;A/{\mathrm{mm}}^{2}$, the zero current zone does not expand much more, which limits the power increase. Based on these explanations, the curves on Figure 2 can be divided into three zones:

Below $100\;A/{\mathrm{mm}}^{2}$, the applied field does not have any impact on the torque as every curves are merged because the bulk is completely penetrated by the magnetic field and provides the same modulation.

Between $100\;A/{\mathrm{mm}}^{2}$ and few thousands of $\;A/{\mathrm{mm}}^{2}$ the bulk are transitioning from complete penetration to a situation where only the edges have current flowing through them.

Eventually, above a certain applied field, the bulks move towards the asymptote of the perfectly diamagnetic bulk. In the two first zones, the machine torque is strongly linked to the bulk critical current while linked only to the applied field in the third zone.

Based on the results of these simulations, we can conclude that even though the $J_{c}$ has an important impact on the performances, it is not necessarily useful to try to go beyond few thousands of $\;A/{\mathrm{mm}}^{2}$. As the threshold above which the bulk has a behavior close to a perfect diamagnetic material depends on the applied magnetic field, we can define a modulation quality factor $F_{Q}$ as the ratio of $B_{1}$ for a given $J_{c}$ and $B_{1}$ for a perfect diamagnetic bulk:(6)$$F_{Q}\left(J_{c}\right)=\frac{B_{1}\left(J_{c}\right)}{B_{1}\left(\infty \right)}$$

Figure 5 shows the evolution of the modulation quality factor for different applied magnetic fields. If we consider a bulk as behaving as a diamagnetic material when its quality factor is above 0.95, it can be observed that $800\;A/{\mathrm{mm}}^{2}$ is required for 1T applied, while $3000\;A/{\mathrm{mm}}^{2}$ is needed to properly screen 4T.

As said above, the machine power is directly linked to the modulation amplitude, itself linked to the critical current density on the edge of the bulk as well as the applied magnetic field. This highlight two essential parameters for choosing the right material to be used for the shielding. Because of different variations of the critical current density with the applied magnetic field, the screening ability of a material also changes with the applied field. In this scope, the model used previously is completed by integrating the variation of the $J_{c}$ with the magnetic field in the E−J law. Figure 6 presents the $B_{1}$ function of the applied magnetic field for a 80mm in diameter disk-shaped bulk made of different superconducting materials presented in the introduction compared with a perfectly diamagnetic bulk.

As expected, the YBaCuO bulk [45] exhibits properties close to a perfect diamagnet for magnetic fields as high as 2T due to a critical current of $4500\;A/{\mathrm{mm}}^{2}$ and starts to shows a tiny difference at 4T where the critical current reached $5000\;A/{\mathrm{mm}}^{2}$. This increase of critical current is due to the well-known fishtail effect [46].

The BaKFeAs materials also shows a promising flux modulation property as, like for the YBaCuO, it is a single crystal with a critical current staying above $1000\;A/{\mathrm{mm}}^{2}$ at 4T [47] thanks to the fishtail effect. These performances needs to be confirmed for larger samples as only few mm^3^ could be synthesized with the current methods [47,48].

However, for the MgB_2_ [41], KFeSe [33], Bi_2_Sr_2_Ca_2_Cu_3_O_x_ (Bi2223) [49], and CaKFeAs [50], we observe a certain value of the applied field for each material at which the $B_{1}$ is maximized, which implies that for a machine made with these materials, an increase in the HTS coil size can lead to a decrease in the magnetic loading and the machine power. This is caused by the decay of the $J_{c}$ with the magnetic field. For instance, while the critical current at 0T is approximately $500\;A/{\mathrm{mm}}^{2}$ for both KFeSe and Bi2223, the $J_{c}$ of Bi2223 decreases to $100\;A/{\mathrm{mm}}^{2}$ at 2T, while the KFeSe is still at $300\;A/{\mathrm{mm}}^{2}$, which explains the better performance of KFeSe compared to Bi2223. Note that although it is not visible on the Figure 6, the $B_{1}$ of YBaCuO and BaKFeAs also reach a maximum but at much higher magnetic fields because of the fishtail effect. Nevertheless, as magnetic field higher than 4 T are not likely to be used in electrical machines, MgB_2_ and KFeSe are still interesting options as they can provide significant flux modulations.

These measurements highlight another key parameter, which is the value of the external magnetic field generated by the superconducting coil. Where for the MgB${}_{2}$ and the KFeSe this value is selected according to the optimize flux modulation, it is only limited by the capacity of generated high magnetic field with the coils in the case of the YBaCuO and the BaKFeSe materials. Indeed, producing higher magnetic fields means a bigger and so heavier superconducting coil as it is presented in Figure 7. It represents the magnetic field produced by an HTS REBaCuO coil in a flux modulation machine function of the HTS tapes mass required in a coil of 352 mm in diameter and 107 mm in length designed using a 2D asymmetric model presented in [22]. Note that generating a large magnetic field requires a heavier coil, as the critical current density of the HTS tapes decreases with the magnetic field. Thus, the ratio of the coil magnetic field and the coil weight decreases with the generated magnetic field, making the magnetic load increase for a flux modulation machine costly.

## 3. Impact of the Synthesis Process on Performances

The most recent flux modulation prototype presented in [23] was realized using multi-seeded YBaCuO bulks. Figure 8 shows the rotor assembled with five of these bulks. The multi-seeded melt growth method (MSMG) allows creating large REBaCuO bulks in a reasonable frame of time; it has been observed, however, that the superconducting properties along the grain boundaries were inferior to the properties in the single crystals [51].

Based on the measurements realized on this prototype, two 3-D models are used to assess the grain boundaries influence on the machine performances:Single-Grain model (SG): A time-dependent H-formulation model, where the superconductors have homogeneous properties. Its critical current density is fixed to $2000\;A/{\mathrm{mm}}^{2}$. This model is used as a reference and corresponds to a single-seeded bulk.Multi-Grain model (MG): A model considering a significant drop of the $J_{c}$ at the grain boundaries of the bulk. Figure 9 shows one of the prototype’s bulk with the four seeds used and the representation of the MG model with the grain boundaries regions. To simplify, the change of the critical current density is considered sinusoidal over the boundary region. The minimal $J_{c}$ in the frontier is fixed at $46\;A/{\mathrm{mm}}^{2}$ to best fit the measurements.

For both models, a bulk with 80 mm in diameter bulk is exposed to an external magnetic flux density along the *z*-direction rising from 0 to 0.55T. This latter value correspond to a current of 120 A flowing in the superconducting coil. The calculation of the flux modulation is done, on the line shown, 5mm over the surface of the superconductor. This emplacement corresponds to the location of the Hall sensor placed out of the cryostat during the experiment.

The impact of the grain boundaries on the flux modulation can be clearly seen on the Figure 10 which shows the comparison of the SG and MG models with the measured data on the prototype. In order to have an insight of the grain boundaries influence on the machine torque, the fundamental magnetic flux density $B_{1}$ is calculated in Table 2 for each model. Thus, the torque produced with the multi-seeded bulks is 23% lower that which would have been produced with a single-seeded bulk. Additionally, as the MG model is precise enough to predict the $B_{1}$ with a 2% error, it can be assumed that the critical current density around the grain boundaries is reduced by 98% and lead to a significant penetration of the magnetic flux in that zone.

The MG model can be used to extrapolate the multi-seeded bulk behavior for higher applied magnetic fields. Therefore, Figure 11 shows the $B_{1}$ value calculated with the MG and SG models for different magnetic fields applied by the coil. With an external field of 0.2T, there is no difference between the use of a multi-seeded and single-seeded bulk. However, with an external field of 2T, the machine torque is almost reduced by a factor of 2 for the multi-seeded bulks. Consequently, the MSMG method offers a good solution to reduce the price and the manufacturing time of the superconducting screens but is not suitable for flux modulation machines with an high magnetic loading.

These results and Figure 3 highlight the necessity of having a very high critical current density even at relatively high magnetic field, and particularly at the edges of the superconducting bulk as screening currents are concentrated on the edges of the bulk and almost null inside. Therefore, an efficient modulation will be obtained by a bulk with a good homogeneity of the performances all along the circumference of the superconductor. The synthesis method used to prepare the sample play a crucial role here and must be chosen wisely as high power machines have a large volume and consequently require large bulks to make their rotors.

Moreover, Figure 12 presents the different possibles inductors for radial and axial flux modulation machines. While some inductors require easily manufactured bulk shapes, such as cuboids or disks, the ideal shapes for axial flux and radial flux structures are ring segment and tile shapes, respectively, which are more difficult to produce. Indeed, not all production methods are suitable for the production of large samples with complex shapes.

Two major type of superconducting bulks can be synthesized: single crystal and poly-crystal. They are fundamentally different, as the single crystal can be seen as a single macroscopic grain with an homogeneous crystal orientation and a tiny number of structural defects, while a poly-crystal is made of multiple grains sintered together with a random crystal orientation and a large number of structural defect due to misalignment between grains, i.e., the grain boundary. This paper will be focused on the Top Seeded Melt Growth (TSMG) and the Infiltration growth (IG) synthesis techniques for single crystals as well as the classic sintering and the Spark Plasma Sintering (SPS) techniques for poly-crystals.

The TSMG method is widely known for the preparation of centimetric-scale single-crystal REBaCuO superconductors. Indeed, this technique is used for commercial bulks surperconductors [52,53] and shows a promising future as the synthesis size limit is regularly increasing [54]. Furthermore, the critical current density of such superconductors is really high and shows good mechanical properties. However, this technique is really time consuming as it requires 230 h for the crystal growth and 300 h of oxygen annealing for a standard $13\;{\mathrm{cm}}^{3}$ cylinder of bulk GdBaCuO [55] and increase with the size of the sample. Furthermore, the presence of the so-called growth sector boundaries makes the homogeneity almost impossible to achieve as a misalignment is occurring along those boundaries [56]. Even more problematic is the evolution of the superconducting properties along the sample radius as reported in several papers [57,58,59]. For example, Antal et al. [45] present a diminuation of 24% of the $J_{c}$ of two samples distant of only 8 mm along the radius of a commercial YBaCuO bulk at 2 T and 4.2 K. Finally, for very large sample the homogeneity is even harder to achieve as it cannot be control during the preparation process, either during the synthesize or the machining, the consequence is the presence of weak parts where superconductivity is bad. This can be seen in Figure 13 where, for a large commercial ring segment-shaped GdBaCuO bulk, a weak area of superconductivity can be seen during a field cooling magnetization at 77 K.

Another method for preparing large size single crystal is the so-called infiltration-growth technique. This method requires a matrix of precursor materials, a liquid reactive material, and like for the TSMG, a seed on the top of the bulk [60,61,62]. Contrary to the TSMG method, it is easy to control the shape and the density of the precursor matrix and so the characteristics of the prepared superconducting bulk. As an example, by using a very porous starting material, it is even possible to prepare a foam-like sample [63,64,65]. However, this method processes the same drawbacks as TSMG which are a long synthesis time as well as a difficulty for preparing very large samples.

Toward the synthesis of large scale REBaCuO with several seeds, an improvement of the grain boundary superconducting properties can be obtained through superconducting welding technique. The welding of YBaCuO bulks requires a solder material with a low peritectic temperature such as REBaCuO with an heavy rare earth, e.g., Yb, Er [66,67,68,69,70,71], or a YBaCuO with a dopant reducing its peritectic temperature such as Ag [72,73,74]. This method has been shown to be more effective than MSMG [75], while being very fast to realize compared to the time required for the TSMG [76,77]. Figure 14 shows a 10mm×10mm×7.5mm GdBaCuO+Ag sample with a ErBaCuO+Ag weld at the middle and its trapped magnetic field map in liquid nitrogen (LN_2_). The weld was realized along the (110/110) plane with 12 h of crystal growth and 48 h of oxygen annealing. It can be seen on the field map that the welding has no impact on the trapped field showing a good connection between the two soldered parts. However, welding is a process that remains difficult to perform for large samples.

In the other side, poly-crystalline bulks are widely prepared by sintering, a very quick, simple, and cheap method [41,50,78]. This also allow the preparation of very large samples scale, tens of centimeters or even more, and with a totally free shape. However, it is hard to guarantee a good homogeneity of the temperature inside the furnace or a good powder mix during the synthesis or the annealing process. This lack of control and precision may induces the apparition of secondary phases with potentially other magnetic properties which lowers the sample performance [78,79]. Furthermore, a poly-crystal shows problematic grain boundaries, due to misorientation, the presence of defects, impurities, and a low density. The consequence is a low connectivity and so a low critical current as well as poor mechanical properties. Furthermore, this random orientation vanishes the macroscopic anisotropy of the crystal by averaging the local anisotropy of all grains.

One way to dramatically improve the performance of a polycrystaline sample is the use of the so-called Spark Plasma Sintering (SPS). It consists of increase the sample temperature by joule effect, a current is applied though the sample to reach very quickly the target temperature with a very good homogeneity all along the sample. At the same time the sample is pressed between the current lead with the same order of pressure than used for the Hot Pressing technique, whereas hours are needed for sintering a sample using a furnace, only tens of minutes to an hour are needed using the SPS technique. Furthermore, the resulting sample has a very high density, easily above 80% of the theoretical density [31,41] or even above 95% [80], a very good grain connectivity, and a very good homogeneity. The consequence is a huge improvement of the critical current and the mechanical properties to reach a level closed to the one of a single crystal with only a tiny fraction of the preparation time.

## 4. Conclusions

We have seen that different superconducting materials are suitable for flux modulation machine. However, it is clear that the properties of ReBaCuO are superior to its counterparts, especially in terms of high field critical current and critical temperature, thus allowing to reach better performances. Nevertheless, the synthesis of large size and homogeneous REBaCuO bulks is difficult and time-consuming despite the different techniques that could be employed to overcome these issues. Therefore, it might be appropriate to consider other materials such as MgB_2_ or something iron-based as an alternative. Indeed, those materials have reasonable properties and their production requires only sintering which allows to rapidly and cost-effectively produce large sized and complex shaped bulk at low cost compared to REBaCuO.

Moreover, much progress can be achieved through doping and SPS processes to reach similar screening capacities to the REBaCuO for magnetic fields below 2T which would already generate a large magnetic loading in an electrical machine. In addition, KFeSe, MgB_2_, and several Pnitides (such as CaKFeAs) do not include any rare earth material unlike REBaCuO, which is of important environmental, political, and economical concern. 

## Figures and Tables

**Figure 1 materials-14-02847-f001:**
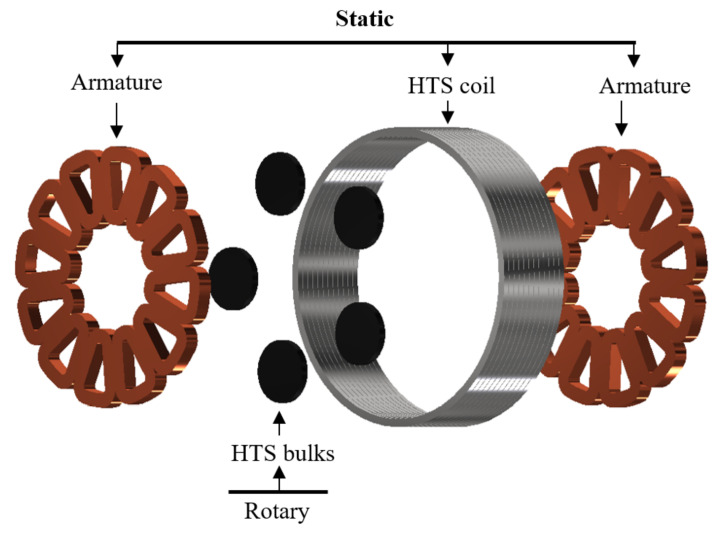
Exploded view of an axial flux modulation machine’s active components. Static parts: two copper armatures and one High-Temperature Superconductor (HTS) coil. Rotating parts: a set of HTS bulks.

**Figure 2 materials-14-02847-f002:**
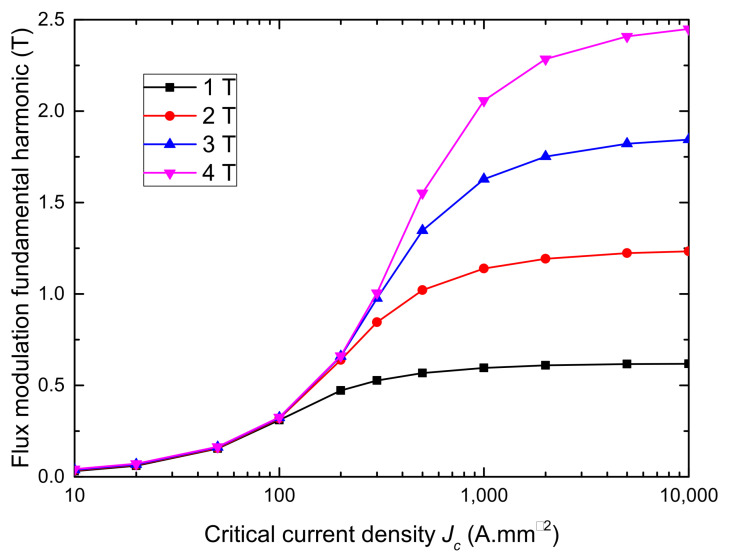
Simulation of the amplitude of the flux modulation fundamental harmonic in the air gap versus the bulk critical current density for applied magnetic fields of 1, 2, 3, and 4T. The machine torque is proportional to the first harmonic of the air gap magnetic field.

**Figure 3 materials-14-02847-f003:**
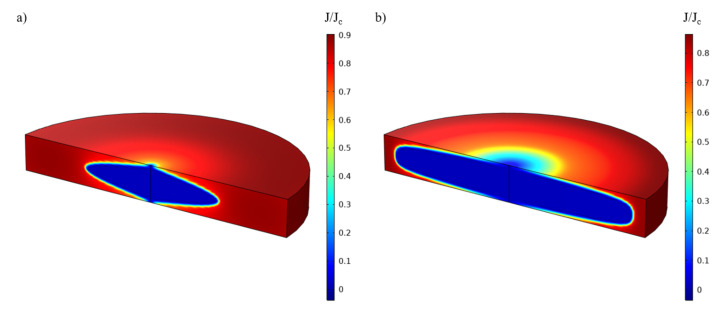
Ratio of current distribution to critical current in a bulk subjected to an applied field of 3T for (**a**) $J_{c}=$
$500\;A/{\mathrm{mm}}^{2}$ and (**b**) $J_{c}=$
$2000\;A/{\mathrm{mm}}^{2}$ simulated using Comsol Multiphysics.

**Figure 4 materials-14-02847-f004:**
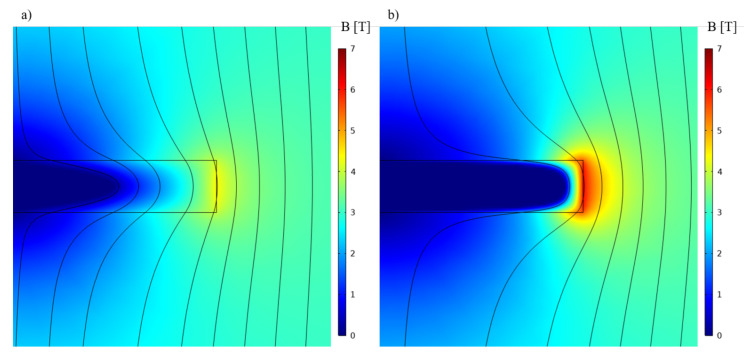
Modulation of the applied magnetic field by the bulk for (**a**) $J_{c}=$
$500\;A/{\mathrm{mm}}^{2}$ and (**b**) $J_{c}=$
$2000\;A/{\mathrm{mm}}^{2}$ simulated using Comsol Multiphysics.

**Figure 5 materials-14-02847-f005:**
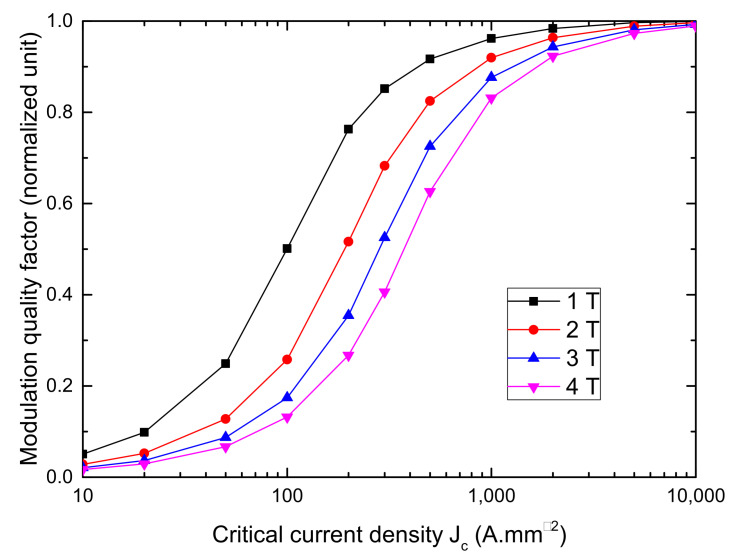
Simulation of the quality of the modulation versus the bulk critical current density for applied fields of 1, 2, 3, and 4 T.

**Figure 6 materials-14-02847-f006:**
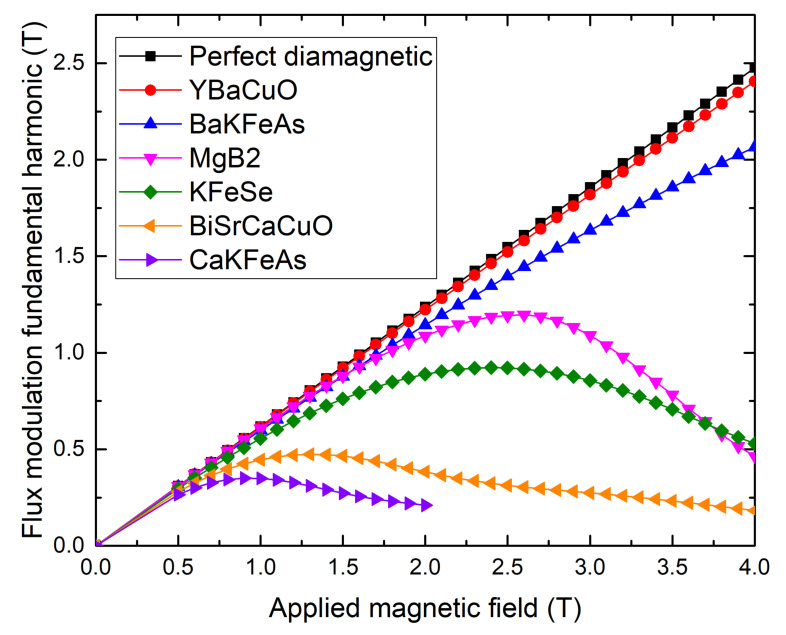
Simulated amplitude of the flux modulation fundamental harmonic in the air gap versus applied magnetic field for different superconducting materials. The material properties are extracted from the works in [33,41,45,47,49,50].

**Figure 7 materials-14-02847-f007:**
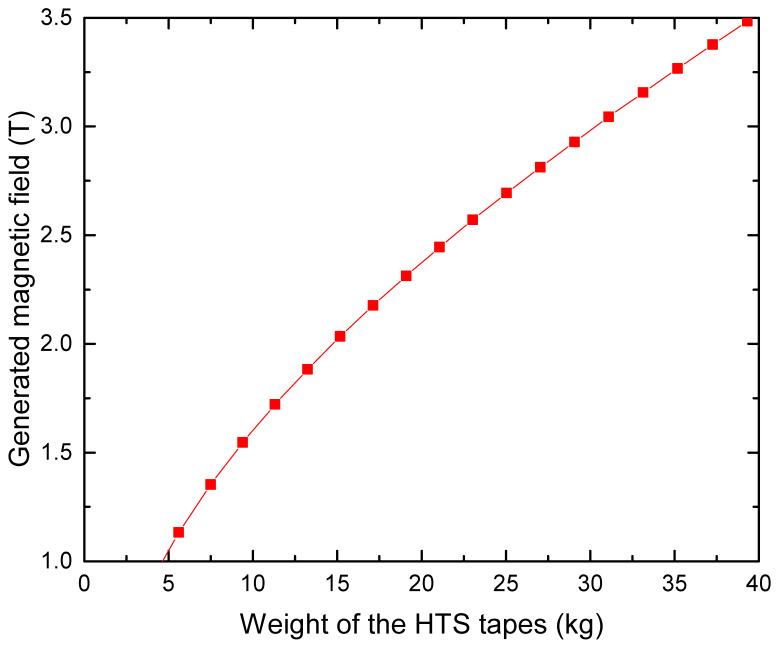
Simulation of the generated magnetic field at the center of a YBCO coil function of its weight for a coil of 352 mm in diameter and 107 mm in length.

**Figure 8 materials-14-02847-f008:**
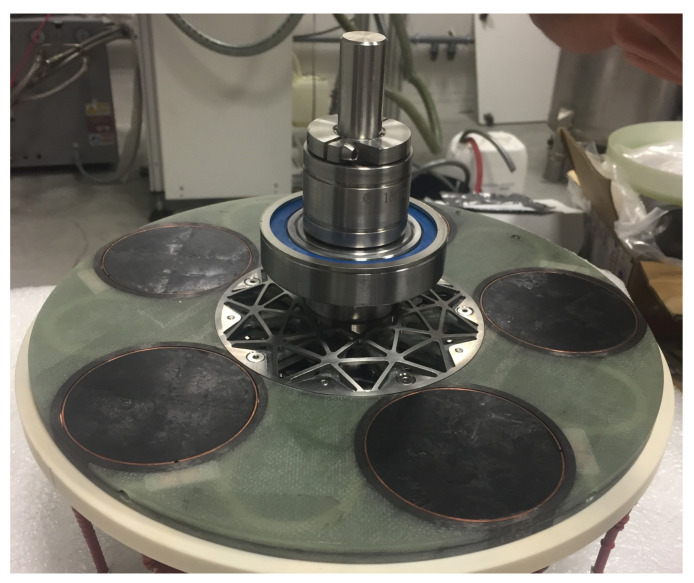
Superconducting rotor of the flux modulation machine prototype.

**Figure 9 materials-14-02847-f009:**
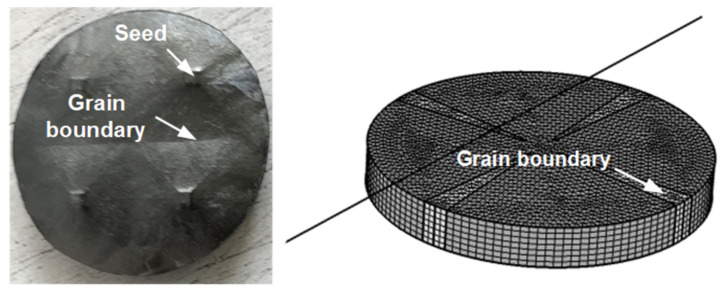
**Left**: Picture of the multi-seeded commercial bulk from ATZ of the 50 kW prototype before the polishing phase. **Right**: Geometry of the MG model considering a reduced critical current density on the grain boundaries.

**Figure 10 materials-14-02847-f010:**
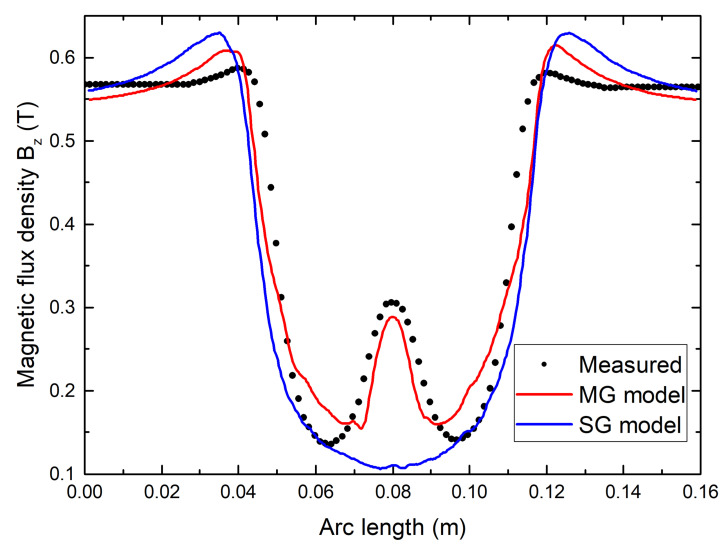
Comparison of the flux modulation calculated by the Single-Grain model (SG) and the Multi-Grain model (MG) with the measured flux modulation in the prototype for an applied magnetic flux density of 0.55T.

**Figure 11 materials-14-02847-f011:**
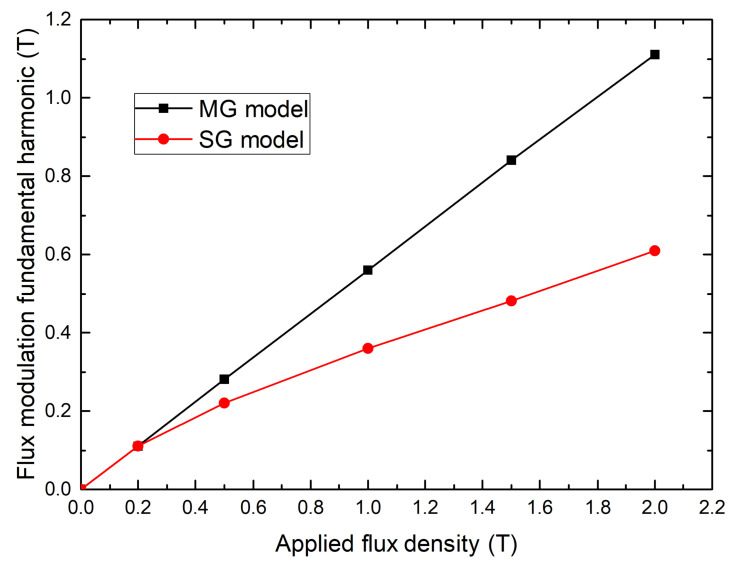
Simulated amplitude of the flux modulation fundamental harmonic in the air gap versus applied magnetic flux density for the SG and MG models.

**Figure 12 materials-14-02847-f012:**
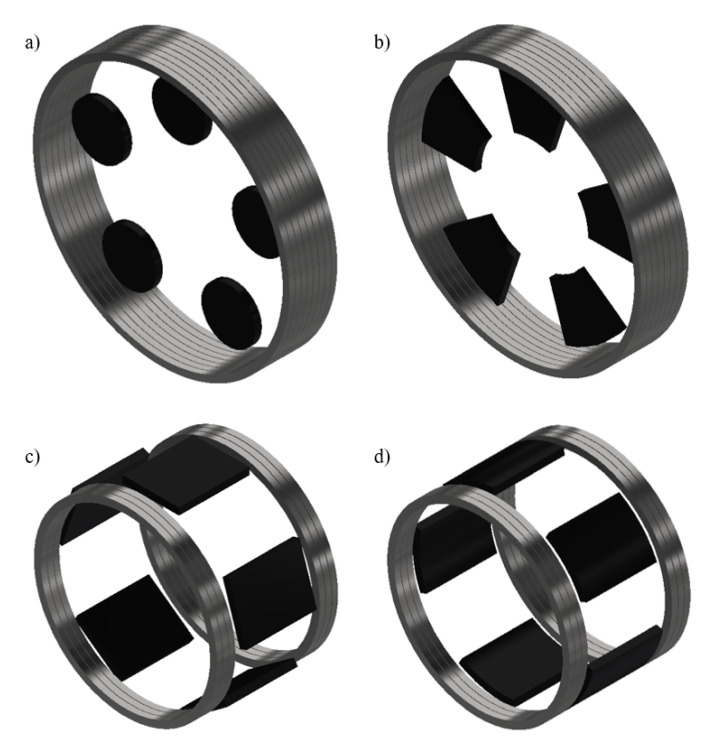
Representation of possible topologies of flux modulation inductors with the different bulk shapes. (**a**) Axial flux topology with disc-shaped bulks. (**b**) Axial flux topology with ring segment-shaped bulks. (**c**) Radial flux topology with cuboid-shaped bulks. (**d**) Radial flux topology with tile-shaped bulks.

**Figure 13 materials-14-02847-f013:**
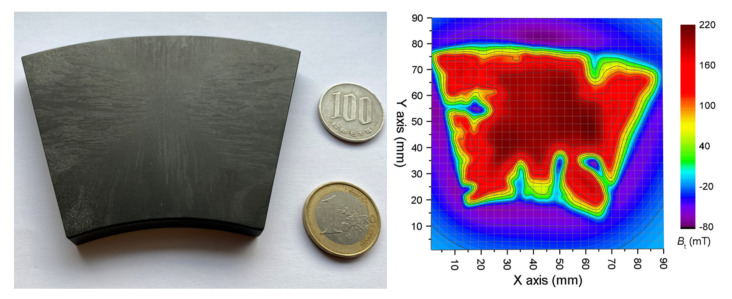
**Left**: Picture of a commercial ring segment shaped GdBaCuO bulk from can superconductors. **Right**: Corresponding measured trapped field map $B_{t}$ of the bulk magnetized by field cooling with a permanent magnet in liquid nitrogen at 77 K.

**Figure 14 materials-14-02847-f014:**
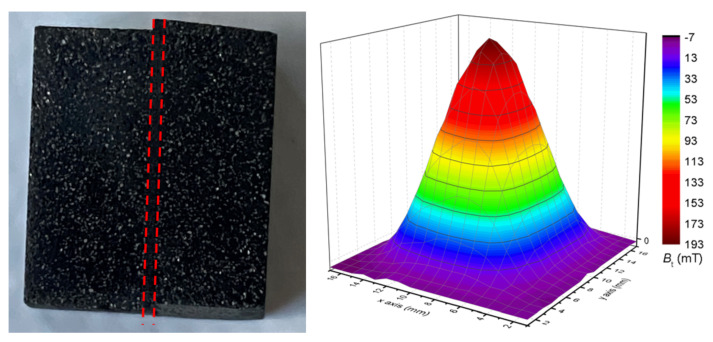
**Left**: Picture of a GdBaCuO commercial bulk produced by Nippon Steel. It has been cut and then welded with an ErBaCuO+Ag welding delimited by the dashed lines. **Right**: Corresponding measured trapped field map $B_{t}$ of the welded bulk magnetized by field cooling with a permanent magnet in liquid nitrogen at 77 K.

**Table 1 materials-14-02847-t001:** Geometrical and physical parameters of the 2-D asymmetric simulation of the bulk on COMSOL.

Symbol	Parameter	Value
$E_{c}$	Critical electric field	1μV/cm
*n*	index of the power law	20
*R*	Bulk radius	40mm
*h*	Bulk thickness	10mm
$\mu_{0}H_{a}$	Applied magnetic field	0–4 T

**Table 2 materials-14-02847-t002:** Comparison of the magnetic flux density fundamental value of the different models with the measured data.

Model	$B_{1}$ (T)
SG model	0.280
MG model	0.220
Measured	0.215

## Data Availability

Data are contained within the article.

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
