# Peer review of "Review on the Use of Superconducting Bulks for Magnetic Screening in Electrical Machines for Aircraft Applications"

_materials, 2021, doi:10.3390/ma14112847_

Round 1

Reviewer 1 Report

The paper by Remi Dorget et al. reports on the simulation and evaluation of the High Temperature Superconductors (HTS) materials, especially about the cuprate oxides. The authors want to clarify what is the most suitable materials for the parts of electrical machines for aircraft applications.
This report consists of two main parts. 
The first one is the simulation of the performance against the critical current (Jc) of the material itself. They found YBaCuO and BaKFeAs are good candidates. In addition, MgB2 and KFeSe are also good for the purpose when the magnetic field is enough low.
In the second part, the authors discuss the realistic preparation of the materials using their simulation of the multi-grain effect based on the experimentally reported result. They conclude that single grain is good for their purpose. In addition, they discuss several methods for growing the sample and their merit and demerit.
In the conclusion section, the authors summarize their discussion and suggest the demerit of selecting cuprate oxide materials and the merit of selecting other materials, KFeSe, MgB2, and several Pnitides.

The referee thinks their report is valuable for readers in the field of electrical machines. However, the referee cannot recommend this paper to be accepted for publication in "Materials" at present. This manuscript should be modified considering the points below.

1) Referee often wondered if this was an experimental or calculated result. It may be due to the lack of citation, especially in the figures. Please add the citation in the figure captions.
2) The referee cannot find the thickness of the disk in their simulation. The authors discuss the penetration depth of the magnetic field, then the referee thinks it must depend on the thickness of the disk.
3) In l. 133, the referee thinks "BaKFeAs" is the typo of the "KFeSe" and "4T" is the typo of "1T". The referee apologizes if this is a misunderstanding of the referee.

Author Response

The paper by Remi Dorget et al. reports on the simulation and evaluation of the High Temperature Superconductors (HTS) materials, especially about the cuprate oxides. The authors want to clarify what is the most suitable materials for the parts of electrical machines for aircraft applications.

This report consists of two main parts.

The first one is the simulation of the performance against the critical current (Jc) of the material itself. They found YBaCuO and BaKFeAs are good candidates. In addition, MgB2 and KFeSe are also good for the purpose when the magnetic field is enough low.

In the second part, the authors discuss the realistic preparation of the materials using their simulation of the multi-grain effect based on the experimentally reported result. They conclude that single grain is good for their purpose. In addition, they discuss several methods for growing the sample and their merit and demerit.

In the conclusion section, the authors summarize their discussion and suggest the demerit of selecting cuprate oxide materials and the merit of selecting other materials, KFeSe, MgB2, and several Pnitides.

The referee thinks their report is valuable for readers in the field of electrical machines. However, the referee cannot recommend this paper to be accepted for publication in "Materials" at present. This manuscript should be modified considering the points below.

1) Referee often wondered if this was an experimental or calculated result. It may be due to the lack of citation, especially in the figures. Please add the citation in the figure captions.

    Answer: The captions of figures showing measured data have been modified accordingly.

2) The referee cannot find the thickness of the disk in their simulation. The authors discuss the penetration depth of the magnetic field, then the referee thinks it must depend on the thickness of the disk.

    Answer: A table listing the geometrical and physical parameters of the bulk has been added at the beginning of the section 2.

3) In l. 133, the referee thinks "BaKFeAs" is the typo of the "KFeSe" and "4T" is the typo of "1T". The referee apologizes if this is a misunderstanding of the referee.

    Answer: Indeed, we wanted to refer to "KFeSe" and the error has been corrected. However, "4T" wasn't a typo error as 1 T is the usual maximum value of the air gap flux density in conventional machines but for superconducting one’s high fields can be considered.

Reviewer 2 Report

The manuscript is well-written, and findings are well-discussed. Authors focused on a very interesting topic, as I believe cryo-electrification like this will be the dominant option for propulsion systems in future electric aircraft. Both academic researchers in applied superconductivity, and superconductor manufacturers would be interested in the content of this article.

Author would need to address the following comments in the revised version of the manuscript:

  1. There are typos and grammatical errors in the text, but it is a minor issue. Please do another proof read (possibly by a native speaker) before submitting the revised version. Examples:

Line 3 in Abstract: remove "the".

Line 24: it should be "illustrated in" instead of "illustrated on".

Line 26: change "an" to "a".

Line 115: change "closed" to "close".

Line 117: change "4 Twhere" to "4 T where".

Line 169: change "fundamental of the magnetic" to "fundamental magnetic".

Line 181: change "a factor 2" to "a factor of 2".

Line 221: change "ether" to "either".

Line 272: remove "Along this contribution".

Line 298: change "heaviest" to "heavier".

…..

  1. There are two relevant papers to the topic of this manuscript, which I think that authors have missed them to bring in literature review section:
  • M Yazdani-Asrami, M Zhang, W Yuan, Challenges for developing high temperature superconducting ring magnets for rotating electric machine applications in future electric aircrafts, Journal of Magnetism and Magnetic Materials 522, 167543, 2021.
  1. At the end of paragraph 1 of Introduction, please mention the NASA target for specific power density of electric propulsion systems, i.e. 20 kW/Kg.
  2. Knowing that future electric aircraft will use liquid hydrogen one way or another, it will bring a huge opportunity to integrate MgB2 superconductors into not only propulsion machines but also whole powertrain/system. How do you see the trend for using MgB2 bulks in machine? Why did not you discuss it in your manuscript? Adding some more paragraphs to manuscript to specifically elaborate it will help readers to know the future trend.
  3. In section 2, please express the main formula of H-formulations for non-expert readers. Would be good to help them to reproduce the results.
  4. Please list in a table, the specifications of the HTS bulk which you have used for your modelling study. It should be shown in a table in the beginning of section 2.
  5. Please express "modulation quality factor" as a formula in the text.
  6. Keep the figure plotting style consistent in the manuscript. Look at the Fig. 5, Y-column. Please use the "." for writing decimal numbers. In addition, the unit of Jc in Fig. 5 is incorrect. It needs to be "-2" in power instead of "2".
  7. Line 161: How this z-direction magnetic field did rise from 0 to 0.55 T? More details are needed.
  8. Add more details to caption of Fig. 9. Comparison of what?
  9. What is the source for Fig. 12? If it is provided by manufacturer, please cite the reference. If it is your own findings, then more details would be needed. Same applies to Fig. 13.
  10. Could you please elaborate more about the effect of electric machine rotating speed on the performance of the bulk in machine? As we know, the NASA target for electric aircraft is to work at high speed, how would this high speed (possibly above 24 krpm) would impact the bulk materials, knowing that they are vulnerable to cracks? Please add a paragraph into the manuscript to explain it.
  11. No need for having appendix as it is very short. Please place Figure A1 into the main text body.
  12. I suggest the type of article to be set as "review" article. Thus, I suggest authors change the title accordingly.

Author Response

The manuscript is well-written, and findings are well-discussed. Authors focused on a very interesting topic, as I believe cryo-electrification like this will be the dominant option for propulsion systems in future electric aircraft. Both academic researchers in applied superconductivity, and superconductor manufacturers would be interested in the content of this article.

Author would need to address the following comments in the revised version of the manuscript:

There are typos and grammatical errors in the text, but it is a minor issue. Please do another proof read (possibly by a native speaker) before submitting the revised version. Examples:

Line 3 in Abstract: remove "the".

Line 24: it should be "illustrated in" instead of "illustrated on".

Line 26: change "an" to "a".

Line 115: change "closed" to "close".

Line 117: change "4 Twhere" to "4 T where".

Line 169: change "fundamental of the magnetic" to "fundamental magnetic".

Line 181: change "a factor 2" to "a factor of 2".

Line 221: change "ether" to "either".

Line 272: remove "Along this contribution".

Line 298: change "heaviest" to "heavier".

…..

    Answer: The typos are removed and the papers was read one more time in order to find and correct remaining typos

There are two relevant papers to the topic of this manuscript, which I think that authors have missed them to bring in literature review section:

M Yazdani-Asrami, M Zhang, W Yuan, Challenges for developing high temperature superconducting ring magnets for rotating electric machine applications in future electric aircrafts, Journal of Magnetism and Magnetic Materials 522, 167543, 2021.

    Answer: The proposed paper is indeed very relevant to the topic. It has been added to the introduction section.

At the end of paragraph 1 of Introduction, please mention the NASA target for specific power density of electric propulsion systems, i.e. 20 kW/Kg.

    Answer: We could not find any reference clearly mentioning 20 kW/kg as a NASA target. We added the value of >10 kW/kg corresponding to the cited book from 2016. Moreover, recent papers (Corduan, Filipenko,...) are using the N-3X goal of 12.7 kW/kg instead.

Knowing that future electric aircraft will use liquid hydrogen one way or another, it will bring a huge opportunity to integrate MgB2 superconductors into not only propulsion machines but also whole powertrain/system. How do you see the trend for using MgB2 bulks in machine? Why did not you discuss it in your manuscript? Adding some more paragraphs to manuscript to specifically elaborate it will help readers to know the future trend.

    Answer: The use of LH2 as main coolant is indeed a huge opportunity for MgB2 as it is one the lightest widespread superconductor. In this article we are focused mainly on bulk and the potential given by the SPS method is very interesting for us since the bulks is the cornerstone of flux modulation machines efficiency. For other components of an aircraft powertrain, the MgB2 appears to be interesting for two main components. Firstly, superconducting cables using MgB2 could be a very efficient way to transport the electrical energy as the current carrying capability of MgB2 is high in low field applications. Additionally, MgB2 is a much cheaper product than REBaCuO. However, studies would need to be conducted to conclude on the best option between MgB2 and YBCO.

    Secondly, the main potential application of MgB2 for the powertrain appears to be for the armature since the MgB2 is easier to realize in a thin strand form and would then produce less AC losses which are an important concern regarding the high frequency of aircraft electrical machines.

    Although these matters are very interesting issues, it seems to us that there are out of the present scope which is the superconducting bulks and it would require dedicated studies to answer these questions with enough confidence.

In section 2, please express the main formula of H-formulations for non-expert readers. Would be good to help them to reproduce the results. 

    Answer: The equations of the H-formulation as well as the E-J law has been included in the manuscript.

Please list in a table, the specifications of the HTS bulk which you have used for your modelling study. It should be shown in a table in the beginning of section 2.

    Answer: A table listing the simulation parameters has been added.

Please express "modulation quality factor" as a formula in the text.

    Answer: The expression has been added.

Keep the figure plotting style consistent in the manuscript. Look at the Fig. 5, Y-column. Please use the "." for writing decimal numbers. In addition, the unit of Jc in Fig. 5 is incorrect. It needs to be "-2" in power instead of "2".

    Answer: We corrected the typos in figure 5

Line 161: How this z-direction magnetic field did rise from 0 to 0.55 T? More details are needed.

    Answer: The applied field is applied through a ramp whose details are given in the description of the H-formulation model we added according to point 5.

Add more details to caption of Fig. 9. Comparison of what?

    Answer: The caption of this figure has been extended.

What is the source for Fig. 12? If it is provided by manufacturer, please cite the reference. If it is your own findings, then more details would be needed.

Same applies to Fig. 13.

    Answer: Some information was added about the origin of the materials used for this paper

Could you please elaborate more about the effect of electric machine rotating speed on the performance of the bulk in machine? As we know, the NASA target for electric aircraft is to work at high speed, how would this high speed (possibly above 24 krpm) would impact the bulk materials, knowing that they are vulnerable to cracks? Please add a paragraph into the manuscript to explain it.

    Answer: In our forecast, the rotational speed of conventional machine is expected to reach speed superiors to 10 krpm as it is one of the main ways to increase the specific power of the machine. However, as the superconducting technology allows to increase the PtM through other means using the high current carrying capabilities, we intend to use the superconducting motors at speeds around 3000 rpm only. It is also consistent with the two following studies from Rolls-Royce:

    https://doi.org/10.3390/aerospace7080107 and https://doi.org/10.1088/1361-6668/ab7779

    In their results, the conventional machine speed reaches 17.5 krpm and requires a 450 kg gearbox while the superconducting counterpart has a rotational speed of 3.5 krpm and no gearbox required.

    Thus, in our case we didn't make any studies so far on the bulk resistance to the centrifugal forces. Yet, our current and next prototypes have working speeds of 5000 rpm and no issues has been observed so far on the bulks.

    These notes have been summarized and added in the article's introduction.

No need for having appendix as it is very short. Please place Figure A1 into the main text body.

    Answer: We removed the appendix A and them put the figure directly in the main text

I suggest the type of article to be set as "review" article. Thus, I suggest authors change the title accordingly.

    Answer: Yes, we agree, the title was modified in consequences.

Round 2

Reviewer 1 Report

The authors succeeded in responding all the referee's comments, then the referee recommends this manuscript will be published in Materials.

Author Response

Thank you for your review,

Respectfully,

Quentin NOUAILHETAS

Reviewer 2 Report

Dear Authors,

Thanks for addressing most my comments, and responding to most my questions.

There is only one response that looks unsatisfactory for me. Certainly the target for power density is 20 kW/Kg for propulsion systems in electric aircraft for MW scale machines. You can find it this number in many references including these two as well as NASA roadmap document. 

10.1109/TTE.2021.3068928

10.1016/j.jmmm.2020.167543

Please correct the number in the end of first paragraph.

regards,

Author Response

The end of the first paragraph has been modified consequently to the reviewer's comment (in red in the text) and corresponding references were added

Respectfully,

Quentin NOUAILHETAS